# Utilizing Sensory and Visual Data in the Value Estimation of Extra Virgin Olive Oil

**DOI:** 10.3390/foods13182904

**Published:** 2024-09-13

**Authors:** Seidi Suurmets, Jesper Clement, Simone Piras, Carla Barlagne, Matilde Tura, Noureddine Mokhtari, Chokri Thabet

**Affiliations:** 1Department of Marketing, Copenhagen Business School, 2000 Frederiksberg, Denmark; ssu.marktg@cbs.dk; 2Social, Economic, and Geographical Sciences Department, The James Hutton Institute, Craigiebuckler, Aberdeen AB15 8QH, UK; simone.piras@hutton.ac.uk (S.P.); carla.barlagne@inrae.fr (C.B.); 3INRAE, UR ASTRO, F-97170 Petit-Bourg, France; 4Department of Agricultural and Food Sciences, Alma Mater Studiorum—Università di Bologna, Viale Fanin 40, 40127 Bologna, Italy; matilde.tura2@unibo.it; 5Department of Agricultural Economics, National School of Agriculture, Meknes 50001, Morocco; nmokhtari@enameknes.ac.ma; 6Institut Supérieur Agronomique Chott Mériem, University of Sousse, Sousse 4042, Tunisia; cthabet@gmail.com

**Keywords:** consumer value, information processing, package design, willingness to pay, olive oil, Morocco, Tunisia

## Abstract

Food evaluation is a topic central to consumer research and food marketing. However, there is little consensus regarding how consumers combine sensory stimuli, product information, and visual impressions to shape their evaluation. Moreover, the bulk of research relies on studies based on questionnaires and declarative responses, raising questions about subliminal processes and their hierarchy in an evaluation process. To address this gap in the literature, we conducted a study with more than 400 participants in Morocco and Tunisia and investigated how factors such as flavor/taste, product information, and packaging design in a variety of olive oils influence visual attention and are reflected in willingness to pay (WTP). We implemented incentivization through an auction to reduce the hypothetical bias in stated WTP values. The results revealed that, compared to tasting the oils, the provision of cognitive information led to an increase in consumers’ WTP. However, a drastic increase in WTP occurred when the consumers were exposed to package designs, overshadowing the formerly dominant effects of product attributes. These findings support theories suggesting a visual perceptual processing advantage due to the picture superiority effect–a picture says a thousand words. Further, it underlines the importance of graphic design in food marketing. The findings have ramifications for food marketing, product development, and pricing strategies.

## 1. Introduction

Balancing the attributes of a food product against willingness to pay (WTP) has long been seen as a consumer’s way to make the most optimal choice [1]. Katt and Meixner [2] outlined several potential attributes as drivers for the WTP in their review, emphasizing that it is still difficult to explain the true relationship between these attributes and what a customer is willing to pay for. Food product attributes range from objective information such as quantity, expiry date, or product type, to information that needs interpretation, which includes brand reputation, environmental issues, or health benefits. The purpose of this research is to gain insights into how people perceive and combine different types of stimuli about a food product’s attributes in their evaluation.

For packed food, many of these product attributes are given through the packaging, which complicates our understanding of how consumers make use of the given attributes in the evaluation of the product. Food attribute information displayed on the packaging is diverse and ranges from readable text to symbolic graphic design elements open for individual interpretation [3,4]. Further, packed food products can be new on the market or well known to the consumer, emphasizing that familiarity with both packaging design and the taste of the food will influence interpretation and evaluation of the food quality and finally determine the WTP [5].

This research aims to understand how food attributes presented on packaging as either readable information or graphic symbolic elements influence both the taste experience and the evaluation of a food product measured in WTP. Yet, a subjective evaluation of a food product is not straightforward to investigate, as both taste perception and information processing are influenced by non-conscious processes that are difficult to express verbally [6]. Therefore, a research protocol based on biometric data was used in a study involving more than 400 participants, who tasted and interpreted product information from various extra virgin olive oils.

## 2. Literature—Theoretical Foundation

### 2.1. Food Information

Comprehensive labeling might give some clarity and certainty for the consumer [7], although packaging labels are often overwhelming or excessively technical, and consumers may have difficulties interpreting them correctly [8]. Research has shown an ambiguity in the interpretation of label information with an impact on a specific evaluation [9].

Information given on packaging about the production site, localness, or nationality has been found to influence food assessment positively [10,11] and specifically consumers’ willingness to pay [12]. Yet, instances in the literature also describe an uncertainty in consumers’ perception of what characterizes a national or a local food product or what constitutes these values [13].

The ambiguity in interpreting packaging labels has also been related to situations where consumers are caught in conflict between given health information and the taste [14,15]. As people in general dislike bitter food, it can be difficult to accept that it might be healthy. A study by Kunz et al. [16] found that people assess health and taste in relation to the situation in which they find themselves, and the context in which the health information is given can alter people’s beliefs and preferences when they taste the food. Context is key in the paper by Kunz et al. [16], especially eating places, leaving the question open as to whether health information given on food packaging would have a similar effect.

Being caught between the interpretation of food information and the taste of the food is described as the cognitive dissonance in food evaluation [17]. Cognitive dissonance is, in general, described as a mental discomfort that causes an increase in cognitive load [18], and it is an unpleasant emotional state that people typically seek to avoid. This study investigates the conflicting experiences between food information provided on the packaging and the taste and examines how such discomfort may negatively influence the evaluation process and WTP.

### 2.2. Packaging Design

Packed food sold in shops (physical or digital) faces the challenge of a balanced interaction between the subjective interpretation of the information provided on the packaging and previous experience. Visual design features on the packaging are, in the marketing literature, described as being core to communicating product attributes and a way to influence taste interpretation [19], and several studies have investigated how different design elements such as color, typeface, illustrations, or shape can impact product judgement [20]. Favier et al. [21] found that a simple design causes the brand to be interpreted as more modern, reliable, and authentic, whereas a complex design gives the impression of a sophisticated brand with high status. However, it has also been found that one packaging design does not have the same impact on all consumers and their WTP [22].

For a healthy food choice, text information about essential food ingredients is seen as the way to support healthy consumption, although the design of nutrition labels and information lacks clarity and needs improvement to be optimal for in-store purchases [23]. Chen and Antonelli [24] argue for a multidisciplinary approach to understand how product information provided through design elements on the front or through a nutrition label on the back of the packaging is interpreted.

However, there is a gap in marketing research regarding the multidimensional approach to the interpretation of packaging design and food tasting, which is a deficiency in marketing research [25]. Many sensory and taste studies have tended to overlook the visual impact of packaging design, and food attributes are in the domain of research related to a verbal or textual description [26]. In their review, the authors (ibid.) outline four main categories that researchers in the domain of tasting tend to focus on, and they conclude that communication about a food product and consumer response is largely driven by the way it is communicated in words and text. In the same review (ibid.), it is acknowledged that visual stimuli from packaging could influence expectations and even impact the perceived taste of the food, although it is less often reported.

Togawa et al. [27] showed how packaging design and especially the placement of an illustration on the packaging can influence consumers’ expectations of flavor, healthiness, and even consumption quantity. This is then described as a “visual-gustatory correspondence effect”, which refers to a cross-modal approach to food perception. Other studies have also utilized a cross-modal approach showing connections between visual stimuli and the evaluation of food healthiness [28]. The conclusion is that the evaluation of a food packaging design needs a holistic research approach, taking packaging design variables into account and investigating the connection between visual impressions and taste.

Still, it remains unclear how different types of information, whether provided verbally, through text, or via illustration, affect consumers’ evaluation of taste, perception of the product, and their final WTP, and, consequently, this study adopts a cross-modal approach.

### 2.3. Visual and Mental Processing

From a neoclassical economic perspective, food purchasers are seen as profit-maximizing entities [29]. However, this rational view rarely holds in real-world situations, where humans are subject to different kinds of biases and take simple heuristic cues [30]. This is described as the gap between stated and revealed preferences [17,31], and it calls for a new way to investigate cognitive dissonance and visual impact in food evaluation.

Differently from tasting the food product, information about the product is conveyed visually through the print on the packaging, and regardless of whether this information is presented as text or an illustration, it requires visual attention. Gazing at product information and getting the full meaning out of the text or illustration is not a straightforward mental process. The literature describes how people’s visual search is influenced by a high degree of routine and acknowledges that such routine might be interrupted by different types of visual stimuli [32].

Various types of information are presented on food product packaging, making it challenging to identify retrospectively, as people often do not have a clear memory of how the different stimuli influenced their mental process and final evaluation of the product (see, e.g., [33]). Additionally, individuals cannot easily explain the interplay between personal preferences and emotions evoked by the stimuli. In other words, people are not able to articulate how these low-conscious processes have influenced their evaluation and the final decision regarding the product.

Stimuli related to a food product that precede tasting are described as exteroceptive cues, typically influencing judgement prior to a tasting. Interoceptive cues, on the other hand, emerge when a person puts the food in their mouth, and different mental processes are involved in the two situations [26], underscoring that exteroceptive cues set the expectations concerning the food product. These low-level mental processes are affected by emotions, which can be detected through bodily reactions such as eye movements and facial expressions. Research within the field of consumer neuroscience has proven the value of utilizing biometric techniques to record and measure these spontaneous reactions in a non-intrusive way [34,35]. Eye-tracking techniques have revealed the complex correlation between visual attention and choice [36,37], showing that people who spend more time gazing at a specific product also evaluate it higher [24].

### 2.4. Focus on Northern African Consumers and Olive Oil

Although marketing science often assumes that findings are generalizable across studies, national borders, and cultures, the majority of existing research is rooted in high-income, industrialized countries [38]. This geographic bias limits the applicability of theoretical insights, especially given that over 80% of global consumers reside in emerging markets and transitional economies (ibid.). Accordingly, there have been calls for more research on an international basis (e.g., [39]), and specifically in emerging markets, including regions such as Africa [38]. Thus, expanding research efforts into African markets is not only pivotal for testing theories and their underlying mechanisms in diverse socioeconomic settings, but it is also essential for refining generalizations and identifying specific boundary conditions (ibid.).

Olive oil is chosen as the object of this study due to its distinctive qualities, having attributes with potentially conflicting interpretations, which could lead to an interesting case of cognitive dissonance. Lastly, olive oil is an important food product in Northern African countries such as Morocco and Tunisia, with the latter being the second-largest producer of olive oil in the world after Spain [40]. Olive oil is an essential component of the local diet for these two countries [41], and in both countries the taste of olive oil is crucial. Tunisian consumers have a high preference for extra virgin olive oil [42], and the region of origin also plays a significant role for them [43]. The literature shows that consumers are willing to pay for locally produced products [11], and in this case, olive oil might face challenges on the market when communicating its region of origin [9] or information on the packaging about extra virgin oil [10].

Consumption of extra virgin olive oil is perceived to generate health benefits [44] and stem from its high content of polyphenols, which results in a more bitter taste and more pungent mouthfeel sensation [45]. The extant literature, most of which has again focused on developed countries, outlines preferences for extra virgin olive oils with a sweet taste and low bitterness and pungency [46,47,48,49] emphasizing a consumer dilemma between choosing the healthiest or the tastiest.

For Morocco and Tunisia, mixed opinions on bitterness and pungency have been found [50]. The unanswered question is whether providing health information about the product (in this case, extra virgin olive oil); information about the region of origin, in this case, Tunisia and Morocco, can push the sensory evaluation of the product and then influence the final WTP.

## 3. Research Design

A cross-modal study design was chosen to clarify how objective information (text) and the subjective interpretation of different flavors (taste) impact evaluation and decision-making. Biometric data collection aimed to uncover the non-conscious processes, while participants’ WTP was used to measure conscious actions. In order to increase external validity, different types of olive oil packaging (types of design) were tested, and to encourage the participants to indicate a realistic price for the oil, we used the auction method that builds upon the Becker–DeGroot–Marschak (BDM) procedure [22], as adapted by Combris et al. [51] and by Barlagne et al. [11]. In this setup, the auction means that the participants had to purchase the extra virgin olive oil if their declared WTP in a randomly extracted round was higher than a randomly extracted price. Extra virgin olive oil was chosen as the object of the study as it entailed a particular dilemma between taste and information given, thus allowing the combination of variation in taste and flavor, health information, and information about the region of origin. By controlling these independent variables, we were able to test the WTP as the dependent variable.

### Study Setting and Procedure

In each country, the participants were asked to evaluate four national brands of extra virgin olive oil, and these oils were characterized by a combination of two binary attributes: (1) low polyphenol vs. high polyphenol content and (2) locally produced vs. not-locally (but nationally) produced. The local product was characterized as coming from the same region where the experiment was conducted. The sensory profile with regards to pungency and bitterness was evaluated by the Professional Committee of DISTAL (Department of Agricultural and Food Sciences of the Alma Mater Studiorum, Università diBologna, recognized by the Italian Ministry of Agriculture, Food Sovereignty, and Forestry) according to official procedures [52,53]. Table 1 provides an overview of the oils used.

The study design combines tasting and information-giving over five phases. In each phase, the participants had to indicate a positive number for WTP for each olive oil as shown in Figure 1 (with the option of choosing 0 too) one by one on a 17-inch computer screen, and, in each phase, the order of olive oils was randomized. In the sixth and last phase, all four bottles were shown simultaneously.

In phase 1, *sensory evaluation*, the participants were provided with four pieces of bread with different oils and asked to state their initial WTP for each oil. Between each sample, participants cleaned their mouths with clean water.

In phase 2, *visual presentation with information,* participants were first exposed to an introduction to the task and then exposed to images of a generic oil bottle one by one accompanied by information regarding the origin (local vs. not local) and polyphenol content (high vs. low).

In phase 3, *introduction to health benefits*, participants were again exposed to an introduction to the task and given more information about the bitter and pungent taste accompanying oils with high polyphenol content. They were also informed about the positive impact of polyphenols on health. This was again followed by the exposure to the same visual images of bottles as in phase 2, and again one by one.

In phase 4, *tasting and reassessment*, participants were again exposed to an introduction to the task, which combined tasting of the four extra virgin olive oils on a piece of bread and, at the same time, exposition to a visual image of an extra virgin olive oil bottle similar to the images in phases 2 and 3. Between each sample, participants cleaned their mouths with clean water.

In phase 5, *package presentations*, the participants were first exposed to an introduction to the task and then to real photos of each of the oil bottles, both front and back.

Finally, in phase 6, participants were shown all four extra virgin olive oils and asked to choose one. The study design is visualized in Figure 1.

The X in Figure 1 was shown on the screen when participants were tasting the extra virgin olive oil, and there was no time limit. Images with bottles of extra virgin olive oil were exposed to the participants for 6 s. An open-ended cell to enter a number was used for indicating the WTP, which was in the local currency (MAD = Moroccan Dirhams, and THD = Tunisian Dinars) and for one liter of extra virgin olive oil. All introduction text was given in Arabic, and the health information given at phase 3 (here, translated into English) had the text: “*A pungent or bitter taste of olive oil indicates a higher content of polyphenols. Olive oil polyphenols are good for your health. Therefore, healthier olive oils are likely to be bitterer and more pungent”*.

The study was conducted in controlled laboratory conditions at local universities in Meknes, Morocco, and in Sousse, Tunisia. Stratified samples were utilized for recruiting urban consumers in the two cities, and the samples were representative of the cities’ populations in terms of age groups and gender. Potential participants were filtered according to four criteria: having no allergy to olive oil, regular consumption of olive oil, being involved in grocery shopping at least from time to time, and knowing the price of one liter of olive oil with good approximation. The group size was set to a minimum of 200 in each city, which would allow differentiation between the participants in the analysis.

A team of local enumerators was trained in running the study and keeping track of the order of taste samples. When a participant joined the study, they were informed about the procedure, the technique, their right to withdraw at any time, and that their data would be fully anonymized. A session started by calibrating the participant and the eye-tracking software. After data collection, each participant was debriefed to ensure full acceptance of the use of data and paid for their effort with the cost of the oil (if purchased) subtracted from their payment.

During all phases, the participant was eye-tracked, utilizing iMotions software v.9.1 and a Tobii Nano eye-tracker (60 Hz), while Qualtrics was used for recording demographic data and auction data. All data were anonymized, stored securely, and used solely in aggregated form to ensure the safeguard of the participants’ rights. The total number of participants was 440 (230 in Morocco, and 210 in Tunisia), resulting in more than 300,000 data observations across phases and stimuli. Collected data were extracted, preprocessed, and then analyzed with JMP Pro 16 software by SAS.

## 4. Results

### 4.1. WTP of Moroccan Consumers

The first purpose of the study was to test how persons who were informed of the origin and health benefits of an extra virgin olive oil would alter their assessment, which in this case was expressed through WTP. To explore this in the five phases (1 to 5), we first ran a generalized linear mixed-effects model. The participants were included as a random effect to account for intra-subject variability across study phases and choice alternatives.

The model’s detailed fit (R^2^ = 0.657, Adjusted R^2^ = 0.656) demonstrated that the model accounted for a substantial proportion of the variation in WTP. With regards to the fixed effects, the different experimental phases had a strong impact on WTP (F(4, 4332) = 37.123, *p* < 0.001), reflecting changes in participants’ valuation over time and in response to different information given in the five phases. Furthermore, the main effects of the polyphenol content (F(1, 4332) = 106.701, *p* < 0.001) were significant. There was also a significant two-way interaction between phase and polyphenol content (F(4, 4332) = 11.613, *p* < 0.001), indicating that the effect of polyphenol content on WTP varied significantly across different phases. The main effect of origin (F(1, 443) = 1.541, *p* = 0.215) did not reach significance level. The least squares means plot for Morocco is visualized in Figure 2.

Following up on the significant two-way interaction, we ran Tukey’s HSD test, accounting for multiple comparisons, to further analyze the differences in WTP as a response to study phases and polyphenol content. The results revealed that in phases 1 and 2 of the study, no significant differences in WTP were observed between extra virgin olive oils with high vs. low polyphenol content. However, in phases 3 and 4, significant differences in WTP became apparent. Specifically, in phase 3, extra virgin olive oils with high polyphenol content were evaluated significantly higher than those with low polyphenol content, with a difference in WTP of MAD 24.387 (*p* < 0.001).

This pattern was similarly observed in phase 4, where high polyphenol content (compared to low polyphenol content) increased the WTP by MAD 18.828 (*p* < 0.001) compared to low polyphenol content in the same phase. In phase 5, the difference in WTP between oils with high and low polyphenol content reduced to MAD 8.29, but this result was below the level of significance (*p* = 0.053), implying no difference in participants’ WTP. This indicates that visual package information may have mitigated the information about polyphenol content and the corresponding health benefits. Table 2 summarizes the values of WTP in the five phases from Morocco.

### 4.2. WTP for Tunisian Consumers

The same analyses (linear mixed-effects model) were carried out with data from Tunisia, enabling the results to be compared. Again, the model’s detailed fit (R^2^ = 0.693, Adjusted R^2^ = 0.691) demonstrated that the model accounted for a substantial proportion of the variation in WTP. The fixed-effects analysis showed that the experimental phase (F(4, 3964) = 128.777, *p* < 0.001), the polyphenol content (F(1, 3964) = 217.748, *p* < 0.001), and the origin (F(1, 3964) = 7.656, *p* = 0.006) all had a significant effect on WTP. The two-way interaction effects between phase and polyphenol content (F(4, 3964) = 28.298, *p* < 0.001) and phase and origin (F(4, 3964) = 6.017, *p* < 0.001) were significant too. Finally, the three-way interaction between phase, origin, and polyphenol content also exceeded the significance level (F(4, 3694) = 4.91, *p* < 0.001. The least squares means plot for Tunisia is visualized in Figure 3.

To understand the impact of polyphenol content on participants’ WTP in different phases of the study, we ran again Tukey’s HSD test that accounts for multiple comparisons. Similarly to data from Morocco, no significant differences in WTP were observed between extra virgin olive oils with high vs. low polyphenol content in phase 1, where participants merely tasted the oils. In phase 2, the WTP for extra virgin olive oils with high polyphenol content exceeded that of the extra virgin olive oils with low polyphenol content by THD 0.83 (*p* = 0.006). The magnitude of the difference in WTP was higher in phase 3, where the participants were willing to pay on average THD 3.03 (*p* < 0.001) more for extra virgin olive oils with high polyphenol content, compared to extra virgin olive oils with low polyphenol content. The significant difference in WTP was also present in phase 4, where high polyphenol content (compared to low polyphenol content) increased the WTP by THD 2.35 (*p* < 0.001). In phase 5, the difference in WTP was THD 0.36, yet below the significance level (*p* = 0.831), indicating again that visual aspects of the packaging overshadowed the impact of polyphenol content on participants’ WTP. Table 3 summarizes the values in the five phases from Tunisia.

### 4.3. Visual Impact in Phases

In phases 2, 3, and 4, the participants were exposed to visual images of generic olive oil bottles presenting information about the origin and polyphenol content. We were interested in investigating whether the duration of visual attention to different areas of interest (AOI), such as text about the origin and polyphenol content of the extra virgin olive oil, in combination with the characteristics of the oil (including taste, when relevant) and study phases, predicts participants WTP. Accordingly, we ran a linear mixed model (LMM) with restricted maximum likelihood (REML) estimation to assess the degree to which the level of visual attention, measured as Total Fixation Duration (TFD), predicts the participants’ WTP.

The LMM included fixed effects for phase, AOI_Origin (Local vs. Non-local), TFD_Origin, AOI_Polyphenols (High/Low), TFD_Polyphenols, and their respective two-way and three-way interactions. Participants were included as a random effect to account for individual differences in WTP and the correlation structure within subjects across the different phases and oil variants. Based on the data from Morocco, the model’s detailed fit (R^2^ = 0.639, Adjusted R^2^ = 0.636) demonstrated that the model is quite effective at explaining the variability in WTP. The LMM analysis revealed a significant main effect of the study Phase (F(2, 2400) = 9.026, *p* < 0.001), AOI_Polyphenols (F(1, 2399) = 100.319, *p* < 0.001), and TFD_Origin (F(1,2568) = 4.7, *p* = 0.031) on WTP. The significant two-way interaction between Phase and AOI_Polyphenols (F(2, 2397) = 12.084, *p* < 0.001) indicated that the effect of polyphenol content on WTP varied across different phases of the study. Finally, the significant three-way interaction between Phase, AOI_Origin, and TFD_Origin (F(2, 2409) = 4.735, *p* = 0.009) implied that there is a more complex interplay when the three variables are combined. The main effects of AOI_Origin, TFD_polyphenols, and other two- and three-way interactions were not significant. In order to obtain more insights into how the different variables affect participants’ WTP, an assessment of the parameter estimates was performed, as presented in Table 4.

Analysis of the parameter estimates revealed that after controlling for other variables, the estimated WTP for extra virgin olive oils with high polyphenol content was significantly higher than the reference level of low polyphenol content. Furthermore, the estimated WTP in phase 2 was significantly lower than in the reference phase 4. With regards to visual attention, it was found that as the viewing time on AOI_Origin increases by 1 ms, the WTP is estimated to increase by 0.002 MAD. The parameter estimates for two-way interactions revealed that compared to the baseline phase 4, the presence of high polyphenol content in phase 2 led to a significant decrease in WTP, whereas in phase 3 it led to a significant increase. Finally, the parameter estimates for the three-way interaction indicated that participants’ WTP was more strongly affected by their attention to local origin in phase 3 compared to the reference phase.

We ran an identical model on the data from Tunisia, resulting in slightly different results. The model’s fit was considerably higher, with an R^2^ value at 0.793 and Adjusted R^2^ at 0.791. The LMM analysis revealed a significant main effect of the AOI_Polyphenols (F(1, 2195) = 413.998, *p* < 0.001) and AOI_Origin F(1, 2194) = 10.04, *p* = 0.002). While the main effect of phase (F(2, 2200) = 2.617, *p* = 0.073) did not reach significance, the two-way interaction between phase and AOI_Polyphenols F(2, 2195) = 39.176, *p* < 0.001) was significant, indicating that the effect of polyphenol content also varied significantly across phases in this country. None of the main effects of TFD_Origin, TFD_Polyphenols, or the other interactions yielded significance. These results provide an indication that the duration of visual attention on informational elements of the package did not significantly affect participants’ WTP, different from Morocco. To better understand how the different variables affect participants’ WTP, an assessment of the parameter estimates was performed, as presented in Table 5.

An assessment of the parameter estimates revealed that, similarly to Morocco, the estimated WTP for extra virgin olive oils with high polyphenol content was significantly higher than the reference level of low polyphenol content. It was also found that Tunisian consumers exhibited higher WTP for locally produced oils. After controlling for other variables in the model, the estimated WTP in phase 2 was significantly lower than in the reference phase 4. With regards to the two-way interactions, it was found that the presence of high polyphenol content in phase 2 led to a significant decrease in WTP, whereas in phase 3, where consumers had been informed of the link between polyphenols and healthiness, high polyphenol content significantly enhanced WTP as compared to the baseline.

Overall, analysis of the measures for visual attention confirmed the findings presented in the previous section but also revealed some relevant differences between Moroccan and Tunisian consumers.

### 4.4. Visual Impact in Phase 5

In phase 5, the participants were exposed to the extra virgin olive oils in their original packaging design. To better understand consumers’ WTP in response to different oil brands, we used survey data to assess the brand knowledge of participants from Morocco and Tunisia. First, we asked the participants to name three oil brands from the top of their minds, and later had them assess their brand knowledge for the extra virgin olive oils used in the experiment. For the latter, we used a 4-point Likert scale (1 = No familiarity at all, 2 = I heard about it, 3 = Quite familiar, 4 = Very familiar). The results covering the average familiarity scores and standard deviations are summarized in Table 6.

As previously mentioned, in phase 5, the participants were exposed to actual product packaging, giving us the opportunity to analyze visual attention for three AOIs: the front, the back, and the brand/logo/name. The previous analysis of WTP across different study phases revealed a higher WTP for extra virgin olive oils with high polyphenol content. Accordingly, we ran a LMM with REML estimation in order to investigate which variables predicted WTP in phase 5. The model included fixed effects for oil alternatives (A, B, C, and D), TFD_Front, TFD_Back, and TFD_Brand, their respective two-way interactions, and participants as a random effect.

Starting with the data from Morocco, the model’s detailed fit (R^2^ = 0.929, Adjusted R^2^ = 0.928) demonstrated that the model was highly effective in explaining the variability in WTP. The LMM analysis revealed a significant main effect of the oil (F(3, 638.5) = 11.613, *p* < 0.001) and TFD_Front (F(1, 787.6) = 6.09, *p* = 0.014). The two-way interaction between the oil and TFD_Back was also significant (F(3, 640.9) = 2.651, *p* = 0.048), as was the interaction between the oil and TFD_Brand (F(3, 646.3) = 4.124, *p* = 0.007).

An assessment of the parameter estimates revealed that the intercept, representing the baseline WTP for oil D (Meissara), was estimated at 121.28, t(511) = 21.82, *p* < 0.001. The parameter estimate for TFD_Front was −0.003, t(787.6) = −2.47, *p* = 0.014, indicating that as the viewing time on the front side of the package increased by 1 ms, the WTP decreased by 0.003. Compared to the baseline of extra virgin olive oil D (Meissara), the parameter estimate for extra virgin olive oil A (El Mallalia) was −2.67, t(639.9) = −2.1, *p* = 0.036, and for extra virgin olive oil C (Volubilia), it was 5.94, t(638.7) = 4.66, *p* < 0.001. This indicates that, overall, extra virgin olive oil C (Volubilia, in a distinctively shaped bottle) was valued significantly higher, and extra virgin olive oil A (El Mallalia) was significantly lower than the baseline oil D (Meissara).

With regards to two-way interactions, the parameter estimate for extra virgin olive oil [A] × (TFD_Back-1043.28) was −0.004, t(641.4) = −2.34, *p* = 0.02, the parameter estimate for extra virgin olive oil [A] × (TFD_Brand-454.97) was 0.01, t(647.2) = 2.78, *p* = 0.006, and the parameter estimate for extra virgin olive oil [C] × (TFD_Brand-454.97) was −0.01, t(648.7) = −2.92, *p* = 0.004. These results indicate that while increased attention to the back of the extra virgin olive oil A had a negative impact on consumers WTP, increased attention to the brand element of sample A led to an increase in WTP. For extra virgin olive oil C, the latter effect was reversed, where increased attention to the brand element led to a decrease in WTP.

An identical model based on data from Tunisia led to slightly different results. The model’s fit was also high (R^2^ = 0.818, Adjusted R^2^ = 0.815), and the fixed effect of extra virgin olive oil was highly significant (F(3, 586.1) = 10.715, *p* < 0.001. Unlike in Morocco, visual attention to the front of the package did not reach the significance level (F(1, 736.5) = 2.825, *p* = 0.093), and neither did any other fixed effects or interactions. With regards to the parameter estimates, the intercept, representing the baseline WTP for extra virgin olive oil D (Ruspina), was estimated at 15.19, t(601.6) = 23.46, *p* < 0.001. The parameter estimate for oil B (Newman’s Own Organic) was 0.57, t(578.7) = 2.72, *p* = 0.007, and for extra virgin olive oil C (Safir) −1.14, t(592.2) = −5.13, *p* < 0.001, indicating that the WTP for extra virgin olive oil B (Newman’s Own Organic) was higher, and the WTP for extra virgin olive oil C (Safir) was lower than the baseline extra virgin olive oil D. These results indicate that while the duration of visual attention effectively predicted WTP among Moroccan consumers, it did not serve as an effective predictor for Tunisian consumers.

### 4.5. Choice Behavior in Phase 6

Previous studies have shown that the chosen product is typically looked at the longest (e.g., [54]). Therefore, we tested whether the TFD on chosen oils is higher compared to non-chosen oils. Thus, we ran a mixed-effects model where selected-or-not was modeled as a fixed effect and participants were included as a random effect.

Based on the data collected in Morocco, the results revealed that the model was significant (F(217, 650) = 1.667, *p* < 0.001). While the main effect of the selection status proved to be highly significant (F(1, 650) = 154.36, *p* < 0.001), the random effect of participants did not yield significance (F(216, 650) = 0.960, *p* = 0.636). The R^2^ value of 0.357 and the Adjusted R^2^ value of 0.143 indicated that the variance explained by the model was moderate. The mean response for the not selected oils was 595.59 ms (SE = 26.355), while for the selected oils it was 1250.46 ms (SE = 45.475). A post-hoc *t*-test with Bonferroni correction revealed that the difference between not selected and selected groups was statistically significant, t(650) = 12.42, *p* < 0.0001. The least squares plot for TFD (with the error bars representing 95% confidence intervals) is presented in Figure 4.

Similar results were obtained for Tunisia, and again the model was significant (F(196, 587) = 2.176, *p* < 0.001), as was the main effect of the selection status (F(1, 587) = 251.146, *p* < 0.001), but not the random effect of participants (F(195, 587) = 0.9, *p* = 0.81). The R^2^ value of 0.421 and Adjusted R^2^ value of 0.227 reflect a moderate proportion of explained variability. The mean response for the not selected oils was 576.25 ms (SE = 27.815), while for the selected oils it was 1457.86 ms (SE = 48.178). A post-hoc *t*-test with Bonferroni correction revealed that the difference between not selected and selected groups was statistically significant, t(587) = 15.85, *p* < 0.0001. The least squares plot for TFD is presented in Figure 5.

## 5. Discussion

We ran a study with six phases, sequentially providing participants with sensory and cognitive information, and assessed how these stimuli affected the evaluation of different extra virgin olive oils, manifested as willingness to pay (WTP). We expected that the amount of information, such as origin and polyphenol content, would impact consumers’ evaluation of extra virgin olive oils.

### 5.1. WTP Based on Tasting and Perceived Information

In the first phase of the study, participants assessed their WTP merely on sensory information. We did not find any significant differences in consumers’ evaluations of different oils among Moroccan or Tunisian consumers. This aligns with the findings of the study by McClure et al. [55], where it was demonstrated that without additional information, consumers were unable to make accurate judgments based on sensory information alone. This first phase sets a foundational understanding that extra virgin olive oil, like many subtly differentiated products, relies heavily on augmented information for value perception.

In the second phase of the study, the participants were informed about the origins and polyphenol contents of the olive oils. While among Moroccan consumers this information failed to have any significant impact on participants’ WTP, Tunisian consumers evaluated oils with high polyphenol content slightly higher. This finding may indicate that labeling practices in Tunisia tend to mention polyphenol content as a positive feature, and in that way Tunisian consumers are more informed about polyphenols.

Prior to the evaluation of the oils in phase 3, the participants were straightforwardly introduced to the health benefits and taste profile of extra virgin olive oils with high polyphenol content. Among both Moroccan and Tunisian consumers, this led to a high-magnitude increase in consumers’ WTP for these extra virgin olive oils. The health-related information, which apparently resonated well with the consumers, enhanced consumers’ perceptions of high-polyphenol extra virgin olive oils and helped them to differentiate between the oils. Furthermore, it implies that consumers in these two countries can alter their evaluation when informed about healthiness, following the findings of Kunz et al. [16].

When participants then tasted the same oils again in phase 4, the WTP for extra virgin olive oils with high polyphenol content remained significantly higher than that for those with low polyphenol content. This can possibly indicate that effective information integration and tasting experience reinforced the value of high polyphenol content. This phenomenon has been studied and explained by Plassmann et al. [56], demonstrating that beliefs and expectations influence subjective experience and consumer value.

### 5.2. Impact from Visual Attention on WTP

In Morocco, the duration of visual attention to the origin information emerged as a significant predictor of WTP, with longer viewing times associated with higher WTP. This finding suggests that when assessing the quality of an extra virgin olive oil, Moroccan consumers placed an interest in gazing at origin information, leading to more cognitive engagement and interpreting it to be relevant, and, subsequently, they increased the WTP for the extra virgin olive oils.

Conversely, in Tunisia, the duration of visual attention did not significantly affect WTP. Yet, Tunisian consumers were also willing to pay higher prices for local oils than for non-local extra virgin olive oils, indicating that the extended viewing times for Tunisian consumers are not necessary to perceive the information about the oil’s origin and what it means in terms of quality, which was also found by Chrysochou et al. [43].

These viewing patterns show differences in information seeking and may further reflect differences in consumer awareness of regional differences and the quality inherent in locally produced products. This is, on the one hand, in line with findings from Barlagne et al. [11], showing that consumers are willing to pay more for locally produced food products, and, on the other hand, it also emphasizes an uncertainty about how to interpret the value related to localness [13].

The duration of visual attention is typically interpreted as an indicator of cognitive processing and increased preference [57,58], which was the case in Morocco. For the Tunisian consumers, the viewing time of origin information was not significantly related to WTP, and, for both groups, the viewing time on polyphenol did not significantly increase WTP. The increase in WTP for extra virgin olive oil with high polyphenol for both groups can, for this study, be attributed to a combined taste experience and the decoding of information.

In phase 5, the participants were presented with photos of the actual product packaging, allowing for a direct evaluation of how visual attention and specific oil packaging characteristics influence evaluation. Differences in viewing time on the branded packaging and its impact on WTP outlined that visual attention is related to factors such as salient design elements and levels of brand familiarity. Interestingly, increased visual attention to the front of bottle C in Morocco impacted the WTP negatively, which could suggest an uncertainty that led to cognitive dissonance [17], and by that, a lower evaluation. Yet, seeing the real bottles increased the WTP for all four olive oils and in both countries.

Overall, the differences between Moroccan and Tunisian consumers in how visual attention influences WTP are notable. Moroccan consumers’ WTP appears to be more sensitive to how they visually engage with product packaging. Tunisian consumers are less influenced by their visual attention, potentially basing their decisions more on pre-existing category perceptions. These observations also speak to the complexities involved in interpreting visual attention in relation to underlying psychological processes, as a longer viewing time can reflect both interest and uncertainty in how to interpret the information.

### 5.3. The Relation to a Final Decision

We expected that an increased amount of information would enhance consumer value perception, resulting in higher WTP, and our results confirmed it. Specifically, in phase 5, where participants were exposed to actual product package designs, there was a notable increase in their WTP. Once the visual packaging was presented, its impact overshadowed the influence of polyphenol content, which had significantly affected consumers’ WTP during the earlier phases of the study. These findings support the premise that the value perceived by consumers is significantly driven by the package design.

For the last phase 6, the study sought to validate if the duration of visual attention on chosen extra virgin olive oils is greater than on non-chosen products. Our findings, both from Morocco and Tunisia, confirmed extended visual attention on selected oils, which could reflect a deeper cognitive processing or a higher interest level in the selected products. This aligns with theories positing that visual attention is both a precursor and an indicator of consumer preferences [28].

## 6. Conclusions and Further Research

In our study, we investigated how sensory stimuli, cognitive information, and visual cues shape consumers’ evaluations of extra virgin olive oil in Morocco and Tunisia. We expected that providing consumers with more information would lead to an increase in subjective product valuation, and hence, higher WTP. A key finding from this study was the significant influence of polyphenol content on consumers’ WTP during the initial phases. However, this influence diminished when consumers were presented with real photos of packaging, which then took precedence, overshadowing the impact of polyphenol content on their WTP. Although we conclude that the core value is in the design of the package, this study did not test the impact of different designs that could explain the variety in viewing patterns and its relation with the evaluation. This calls for further research with a specific focus on different packaging designs for olive oils, able to outline what would increase WTP.

### 6.1. Further Research Related to Cultural Impact

Although findings showed similar trends across cultures, the study also revealed differences across cultural landscapes. With regards to the region of origin, the results among Moroccan consumers showed that visual attention to this information significantly influenced their WTP. Among Tunisian consumers, however, the duration of visual attention did not significantly predict WTP. Tunisian consumers showed preference for local extra virgin olive oils over non-local oils but did not exhibit prolonged visual attention that would have predicted their WTP, implying that immediate information recognition was sufficient to form their evaluation.

One potential explanation for the fast interpretation could be the difference in education levels between the two study samples. While the Moroccan sample closely represented its general population, the Tunisian sample had a large share of university-educated participant. The impact of education on information processing has also been addressed in the literature, namely in a study focusing on participants’ use of heuristic vs. factual cues. Xing and Isaacowitz [59] found that higher education is associated with more attention towards factual rather than heuristic cues, implying a more systematic and fast decision-making approach.

Further, the people recruited for this study had to be familiar with buying, cooking with, and using extra virgin olive oil in their household, which was chosen to ensure that participants could be engaged in the tasks given in the six phases and were not declaring random WTP values. The choice of excluding participants with less familiarity with extra virgin olive oil and novice consumers could affect the generalizability of the findings. These insights obviously call for further research with a specific focus on the relation between cultural, economic, and lifestyle factors that could influence the interpretation of olive oil attributes and how these are cognitively processed.

### 6.2. Further Research Related to Biometric Methods

The results in this study illustrate the complexities of employing biometric measures, such as eye-tracking. As previously discussed, the duration of visual attention is commonly viewed as an indicator of cognitive processing depth and preference. However, it can also signify uncertainty or difficulty in extracting information. For instance, in Morocco, increased attention to the front side of the packaging negatively impacted WTP, which we interpreted as an indication of uncertainty, confusion, or deeper scrutiny and of a lower evaluation of the product as a result. Conversely, during phase 6, the preferred and selected choice alternatives attracted significantly longer viewing times than the non-selected alternatives among both Moroccan and Tunisian consumers, thereby signaling clarity in consumer preference.

It has been argued that the use of eye-tracking is better suited for investigating attention and retention to the upper level of the purchase funnel rather than other stages of the purchase funnel, such as engagement and final purchase [60], which may imply that eye-tracking is primarily useful for gaining insights into initial consumer interest and attention and less effective for the deeper analysis of subsequent decision-making stages, where other factors such as cultural influences, economic considerations, or personal preferences can play a more critical role. This study showed both the benefits and challenges of using eye movement data in all phases of a food evaluation process, and other studies can help develop protocols with biometric data.

Finally, the study highlights the challenges of replicating results. Despite similarities in terms of their culture, the production of extra virgin olive oil, and consumption between Morocco and Tunisia, and despite having an identical study design, the findings related to visual attention varied significantly. These differences could suggest variations in how visual cues are processed or valued due to cultural or demographic factors. It was not within the scope of this study to attribute these differences in viewing behavior to specific factors such as educational levels, cultural differences, or a combination of other variables. Therefore, additional research is needed to explore these aspects more thoroughly.

## Figures and Tables

**Figure 1 foods-13-02904-f001:**
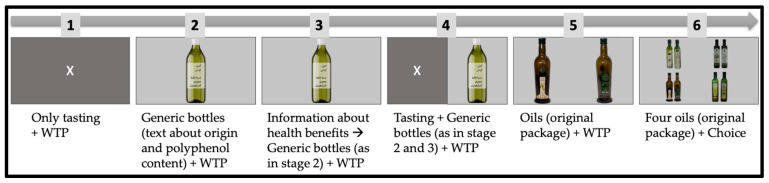
Phases in the study design where the “X” indicates phases with tasting.

**Figure 2 foods-13-02904-f002:**
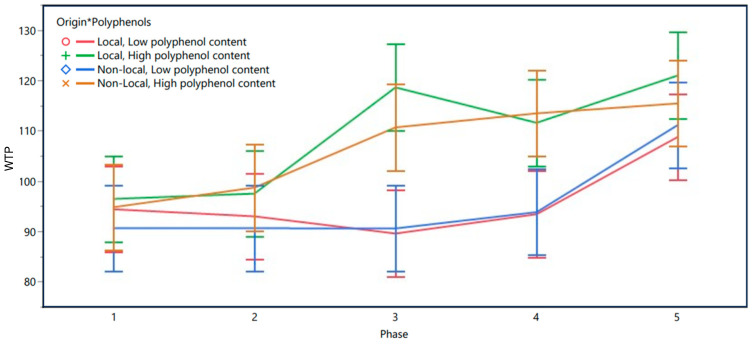
WTP in Morocco as predicted by experimental phases, origin, and polyphenol content.

**Figure 3 foods-13-02904-f003:**
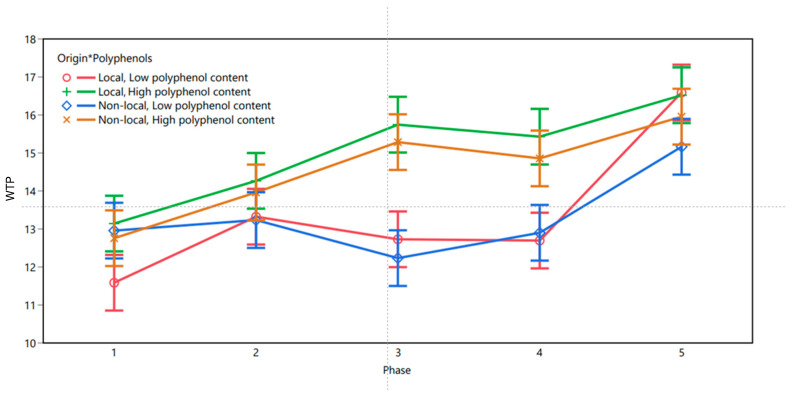
WTP in Tunisia as predicted by experimental phases, origin, and polyphenol content.

**Figure 4 foods-13-02904-f004:**
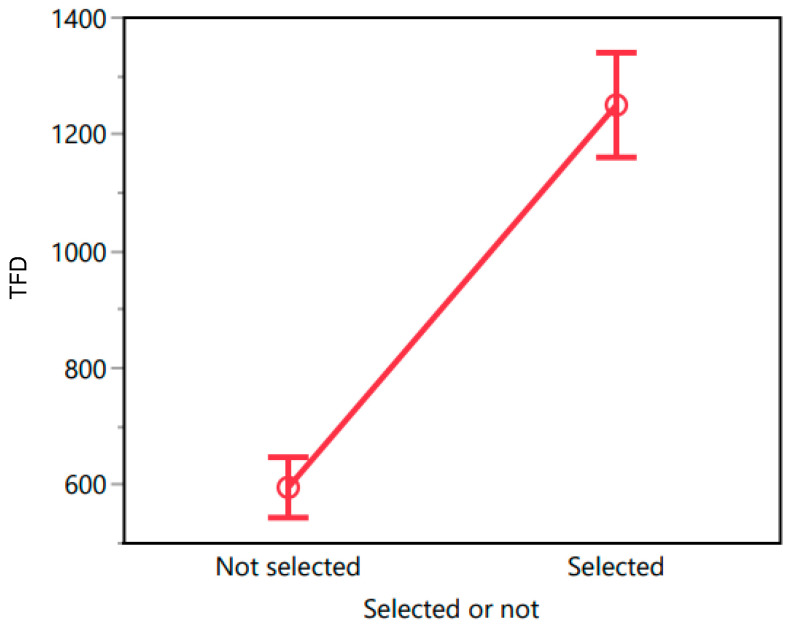
Total fixation duration (TFD) for not selected oils and selected oils in Morocco.

**Figure 5 foods-13-02904-f005:**
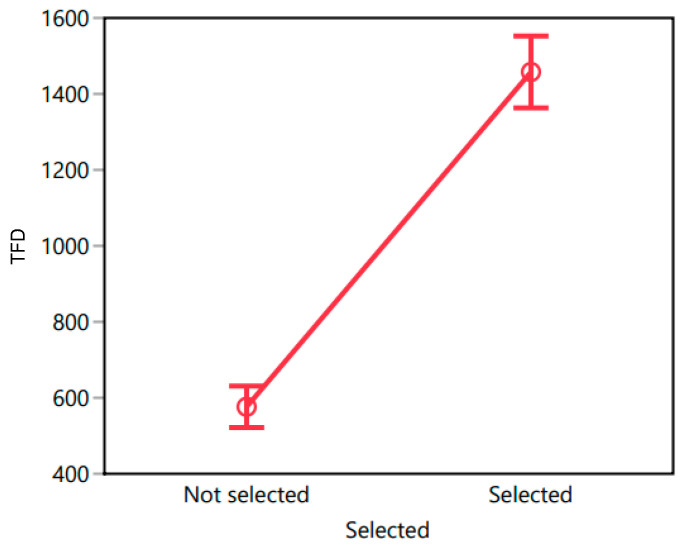
Total fixation duration (TFD) for not selected oils and selected oils in Tunisia.

**Table 1 foods-13-02904-t001:** Extra virgin olive oil samples used in the study.

	Oils Used in Morocco			Oils Used in Tunisia		
	Oil Brand	Origin	Polyphenol Content	Oil Brand	Origin	Polyphenol Content
Oil A	El Mallalia	Not local	Low	Tesoro del Rio	Not local	High
Oil B	Extra Vierge Ouad Ourika	Not local	High	Newman’s Own Organic	Local	High
Oil C	Volubilia	Local	High	Safir Selection	Not local	Low
Oil D	Extra Vierge Meissara	Local	Low	Ruspina Organic	Local	Low

**Table 2 foods-13-02904-t002:** Differences in WTP in Morocco—Tukey’s HSD test.

Phase	Estimate: Low Polyphenols	Std Error	Estimate: High Polyphenols	Std Error	Difference High vs. Low	Lower CL	Upper CL	*p*-Value
1	97.71	3.92	95.81	3.92	3.10	−5.24	11.44	0.976
2	91.93	3.92	98.30	3.92	6.37	−1.99	14.73	0.319
3	90.39	3.92	114.77	3.92	24.39	15.99	32.78	<0.001
4	93.80	3.92	112.63	3.92	18.83	10.49	27.17	<0.001
5	110.07	3.92	118.33	3.92	8.29	−0.06	16.64	0.053

**Table 3 foods-13-02904-t003:** Differences in WTP in Tunisia—Tukey’s HSD test.

Phase	Estimate: Low Polyphenols	St. Error	Estimate: High Polyphenols	Std Error	Difference High vs. Low	Lower CL	Upper CL	*p*-Value
1	12.27	0.339	12.95	0.339	0.679	−0.017	1.374	0.063
2	13.28	0.339	14.11	0.339	0.835	0.139	1.531	0.006
3	12.48	0.339	15.51	0.339	3.035	2.338	3.73	<0.001
4	12.79	0.339	15.14	0.339	2.345	1.65	3.04	<0.001
5	15.87	0.339	16.23	0.339	0.36	−0.337	1.057	0.831

**Table 4 foods-13-02904-t004:** Parameter estimates for phases 2–4, Morocco.

Effect	Estimate	SE	DF	t	*p*-Value
Intercept	96.542	4.436	484.7	21.77	<0.001
Phase 2	−5.024	1.184	2396	−4.24	<0.001
AOI_Polyphenols (High)	8.439	0.872	2399	10.02	<0.001
TFD_Origin	0.002	0.001	2568	2.16	0.031
Phase 2 × AOI_Polyphenols (High)	−5.529	1.183	2396	−4.67	<0.001
Phase 3 × AOI_Polyphenols (High)	4.341	1.189	2397	3.65	<0.001
Phase 3 × AOI_Origin (Local) × TFD_Origin	0.003	0.001	2408	3.08	0.002

**Table 5 foods-13-02904-t005:** Parameter estimates for phases 2–4, Tunisia.

Effect	Estimate	SE	DF	t	*p*-Value
Intercept	14.033	0.352	322.8	39.83	<0.001
Phase 2	−0.159	0.074	2199	−2.15	0.032
AOI_Polyphenols (High)	1.058	0.052	2195	20.35	<0.001
AOI_Origin (Local)	0.164	0.052	2194	3.17	0.002
Phase 2 × AOI_Polyphenols (High)	−0.629	0.073	2195	−8.61	<0.001
Phase 3 × AOI_Polyphenols (High)	0.0462	0.073	2195	6.31	<0.001

**Table 6 foods-13-02904-t006:** Brand knowledge among consumers.

	Oils Used in Morocco		Oils Used in Tunisia	
	Oil Brand	Brand Familiarity (1 = Low/4 = High)	Oil Brand	Brand Familiarity (1 = Low/4 = High)
Oil A	El Mallalia	1.465 (SD = 0.796)	Tesoro del Rio	1.135 (SD = 0.418)
Oil B	Extra Vierge Ouad Ourika	1.3 (SD = 599)	Newman’s Own Organic	1.063 (SD = 0.28)
Oil C	Volubilia	1.804 (SD = 0.971)	Safir Selection	1.553 (SD = 0.76)
Oil D	Extra Vierge Meissara	1.22 (SD = 0.058)	Ruspina Organic	1.534 (SD = 0.789)

## Data Availability

The original contributions presented in the study are included in the article, further inquiries can be directed to the corresponding author.

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
