# Peer review of "Utilizing Sensory and Visual Data in the Value Estimation of Extra Virgin Olive Oil"

_foods, 2024, doi:10.3390/foods13182904_

Round 1

Reviewer 1 Report

Comments and Suggestions for Authors

1.Although the purpose of the study is mentioned towards the end, the introduction fails to provide a concise thesis statement that encapsulates the main argument or hypothesis of the research. A clear thesis statement would help guide the reader's expectations regarding the specific contributions of this study.

2.The introduction jumps between discussing attributes influencing willingness to pay (WTP) and the role of packaging design without clear transitions. This lack of coherent flow may confuse readers regarding how these different elements are interrelated in the context of the study.

3.While terms like "willingness to pay," "objective factors," and "subjective aspects" are mentioned, the introduction does not provide comprehensive definitions or explanations for these key terms. This could lead to ambiguity in the understanding of how these concepts were applied throughout the study.

4.The central research question posed at the end is complex and can benefit from simplification or clarification. The phrase "objective versus subjective interpretive claims" could be more explicitly defined earlier in the introduction, helping readers understand the distinction being made and its relevance to both taste perception and WTP.

5.Several critical theories or concepts, such as cognitive dissonance and visual-gustatory correspondence, have not been sufficiently elaborated. The introduction of these concepts should include a more thorough explanation of their relevance and application in the current research context, allowing readers to understand their significance in depth.

6.The literature review summarizes the findings from multiple studies without synthesizing them to highlight overarching themes or draw clear conclusions that would support the research focus. A synthesis of the findings can reveal trends, contradictions, or gaps in the literature, thus providing a stronger rationale for the current study.

7.Key terms such as "cognitive dissonance," "exteroceptive cues," and "visual-gustatory correspondence" visual-gustatory correspondence are used without a sufficient definition or context. This can lead to misunderstanding and does not allow readers unfamiliar with these concepts to fully understand their implications. Each term should be clearly defined and contextualized within the research framework.

8.The section on Northern African consumers (2.4) presents an important consideration but does not fully develop how cultural factors interact with consumer perceptions of olive oil. A more comprehensive discussion regarding cultural attitudes towards food products, specifically olive oil, would enhance the depth of the analysis. It may also be beneficial to explore how these cultural dimensions affect cognitive dissonance, as described earlier.

9.While the participant selection criteria are outlined, there is little discussion about how the sample size was determined or why it is sufficient to represent local populations in Morocco and Tunisia. Furthermore, the notion of being "representative" could be better explored by considering variables such as socioeconomic status and cultural differences that may influence consumer behavior.

10.The study required participants to have prior knowledge and experience with olive oil, which may introduce selection bias. Individuals who have strong familiarity or positive prior experiences with olive oil might influence the outcome differently than less experienced consumers. Addressing this potential bias and considering how it might affect the generalizability of the findings would strengthen the credibility of the study.

11.The phases in the study design are described, but the transition between phases lacks clarity regarding how participants relate their experiences across activities (e.g., sensory evaluation vs. health benefit information). Explicitly explaining how participants process information and whether they may alter their preferences during the procedure would elucidate the research process.

12.While the section mentions the informed consent process and participants’ right to withdraw, there could be more emphasis on ethical considerations, particularly regarding how sensitive data from biometric tracking will be handled and the safeguards in place to protect participants’ anonymity. Detailed ethical protocols will further demonstrate the research's adherence to the ethical standards.

13.The results section dives into statistical details without adequately contextualizing the findings in relation to the study's objectives. A clearer articulation of how these results specifically address the research questions or hypotheses would enhance readers' comprehension of the significance of these findings.

14. This section uses various terminology interchangeably, such as "WTP," "mean WTP," and "estimated mean WTP," estimated mean WTP, without clarifying these distinctions. A consistent and clear terminology throughout the Results section minimizes confusion and enhances clarity.

15.While differences between Moroccan and Tunisian consumers have been observed, there is limited exploration of the underlying reasons for these differences. A discussion that considers cultural, economic, or lifestyle factors influencing these variations in WTP could add depth to the findings.

16.The results for Moroccan and Tunisian participants are presented separately but would benefit from a comparative analysis section that synthesizes findings across the two groups. Highlighting similarities and differences explicitly enhances the clarity of cross-cultural findings and provides richer insights.

17.The discussion presents numerous findings and insights but lacks a clear organization. The section would benefit from distinct subsections or thematic groupings that systematically address key aspects, such as sensory evaluation, the impact of information, cultural differences, and implications for marketing strategies. This would enhance readability and help guide the reader more effectively through the authors’ interpretations and conclusions.

18. While some prior studies are referenced (e.g., McClure et al., 2004; Plassmann et al., 2008), the discussion largely operates in isolation from existing literature. A more robust integration of the current findings with the relevant literature would help situate the study within a broader context, allowing for a deeper understanding of how these results contribute to existing knowledge in the field.

19.Although the discussion hints at cultural differences in consumer behavior between Moroccan and Tunisian participants, it does not delve deeply into the implications of these differences. A more thorough examination of how cultural perceptions and values influence preferences and WTP can provide valuable insights.

20. The discussion states that "increased visual attention to the front of the package negatively impacted WTP," which is somewhat counterintuitive without a clear explanation. More attention should be paid to explaining why increased attention might signify confusion or uncertainty rather than value. A clear rationale for these findings would enhance the interpretative depth of the discussion.

21.While the conclusion restates some key findings, it does not effectively summarize the main contributions of the study in a clear and concise manner. A more explicit presentation of the primary findings and their significance would help reinforce this study's contributions to the field.

22. The conclusion briefly touches on cultural and demographic factors that may have influenced the findings but does not provide sufficient exploration of how these factors could be systematically examined in future research. Further elaboration of specific cultural influences or demographic characteristics would enhance the depth of the discussion.

23.While the conclusion acknowledges challenges in replicating the results, it does not explicitly outline the limitations of the current study. Identifying and discussing the strengths and weaknesses of the research design, methodological choices, or sample characteristics would provide a more balanced view and increase the credibility of the findings.

24. Although this conclusion calls for additional research to explore unexplained variations in visual attention, it lacks specificity regarding what future studies might investigate. Outlining potential research questions, methodologies, or contexts that could build on these findings would provide clearer directions for subsequent research.

Author Response

1.Although the purpose of the study is mentioned towards the end, the introduction fails to provide a concise thesis statement that encapsulates the main argument or hypothesis of the research. A clear thesis statement would help guide the reader's expectations regarding the specific contributions of this study.
Thanks for your advice. We have revised the introduction and provided a clear statement.  

2.The introduction jumps between discussing attributes influencing willingness to pay (WTP) and the role of packaging design without clear transitions. This lack of coherent flow may confuse readers regarding how these different elements are interrelated in the context of the study.
Thanks for your comment. We revised the introduction and made a better flow, clarifying and outlining the link between food attributes and packaging design.

3.While terms like "willingness to pay," "objective factors," and "subjective aspects" are mentioned, the introduction does not provide comprehensive definitions or explanations for these key terms. This could lead to ambiguity in the understanding of how these concepts were applied throughout the study.
Thanks for your comment. We fully see the problem in presenting concepts in the introduction without a full definition. We have made a radical revision of the introduction.  

4.The central research question posed at the end is complex and can benefit from simplification or clarification. The phrase "objective versus subjective interpretive claims" could be more explicitly defined earlier in the introduction, helping readers understand the distinction being made and its relevance to both taste perception and WTP.
Thanks for your comment and as mentioned above, we have made a radical revision of the introduction.

5.Several critical theories or concepts, such as cognitive dissonance and visual-gustatory correspondence, have not been sufficiently elaborated. The introduction of these concepts should include a more thorough explanation of their relevance and application in the current research context, allowing readers to understand their significance in depth.
Yes, we agree that the connection between theoretical concepts and current research design should be better. We have revised the section to make the link clearer.

6.The literature review summarizes the findings from multiple studies without synthesizing them to highlight overarching themes or draw clear conclusions that would support the research focus. A synthesis of the findings can reveal trends, contradictions, or gaps in the literature, thus providing a stronger rationale for the current study.
Thanks for your comment. We have revised the sub-sections in the literature review (section 2) and outlined better the connection between these concepts and the aim for the study.

7.Key terms such as "cognitive dissonance," "exteroceptive cues," and "visual-gustatory correspondence" visual-gustatory correspondence are used without a sufficient definition or context. This can lead to misunderstanding and does not allow readers unfamiliar with these concepts to fully understand their implications. Each term should be clearly defined and contextualized within the research framework.
Thanks, see our previous comments. The risk of misunderstanding the meaning should now be minimized.

8.The section on Northern African consumers (2.4) presents an important consideration but does not fully develop how cultural factors interact with consumer perceptions of olive oil. A more comprehensive discussion regarding cultural attitudes towards food products, specifically olive oil, would enhance the depth of the analysis. It may also be beneficial to explore how these cultural dimensions affect cognitive dissonance, as described earlier.
Thanks for your comment. We have added a paragraph and made a revision of section 2.4

9.While the participant selection criteria are outlined, there is little discussion about how the sample size was determined or why it is sufficient to represent local populations in Morocco and Tunisia. Furthermore, the notion of being "representative" could be better explored by considering variables such as socioeconomic status and cultural differences that may influence consumer behavior.
Thanks, we have explained more in detail the sample size. We take the discussion about being representative in the final section of the manuscript (a new section 6.1) where it fits better the into the flow.

10.The study required participants to have prior knowledge and experience with olive oil, which may introduce selection bias. Individuals who have strong familiarity or positive prior experiences with olive oil might influence the outcome differently than less experienced consumers. Addressing this potential bias and considering how it might affect the generalizability of the findings would strengthen the credibility of the study.
This is a relevant question to place. We have added a paragraph in the last section of the manuscript and elaborate on this specific topic. Thanks for bringing it up.

11.The phases in the study design are described, but the transition between phases lacks clarity regarding how participants relate their experiences across activities (e.g., sensory evaluation vs. health benefit information). Explicitly explaining how participants process information and whether they may alter their preferences during the procedure would elucidate the research process.
Thanks for your relevant comment. The purpose with the five phases in the study design was to investigate how people process information (eye-tracking) and how the level of information might alter their evaluation/preferences (taste and WTP). We have added some text about the research design to help the reader better understand the setup.

12.While the section mentions the informed consent process and participants’ right to withdraw, there could be more emphasis on ethical considerations, particularly regarding how sensitive data from biometric tracking will be handled and the safeguards in place to protect participants’ anonymity. Detailed ethical protocols will further demonstrate the research's adherence to the ethical standards.
Thanks for this very relevant comment. We have added the ethical statement by the end of the manuscript.

13.The results section dives into statistical details without adequately contextualizing the findings in relation to the study's objectives. A clearer articulation of how these results specifically address the research questions or hypotheses would enhance readers' comprehension of the significance of these findings.
Thanks for your comment. Yes, we agree that a clearer articulation will help the reader to follow the analysis. Using hypotheses could be a sensible way forward, although it is our opinion that it would make other parts that the manuscript less readable and it would not be a way forward for this manuscript.

14.This section uses various terminology interchangeably, such as "WTP," "mean WTP," and "estimated mean WTP," estimated mean WTP, without clarifying these distinctions. A consistent and clear terminology throughout the Results section minimizes confusion and enhances clarity.
Thanks for this comment. We have removed the ambiguity in the word “mean”. We still use the word “estimate” in relation to the LMM analysis.

15.While differences between Moroccan and Tunisian consumers have been observed, there is limited exploration of the underlying reasons for these differences. A discussion that considers cultural, economic, or lifestyle factors influencing these variations in WTP could add depth to the findings.
Thanks for your relevant comment. We have extended this discussion in section 5.

16.The results for Moroccan and Tunisian participants are presented separately but would benefit from a comparative analysis section that synthesizes findings across the two groups. Highlighting similarities and differences explicitly enhances the clarity of cross-cultural findings and provides richer insights.
Thanks for your comment. We mostly consider it to be more readable presenting the results from Morocco and Tunisia separately. We also acknowledge the value in giving the reader reflections about similarities and differences between the two countries. We have made this clearer and added this to the debate in the last sections of the manuscript.

17.The discussion presents numerous findings and insights but lacks a clear organization. The section would benefit from distinct subsections or thematic groupings that systematically address key aspects, such as sensory evaluation, the impact of information, cultural differences, and implications for marketing strategies. This would enhance readability and help guide the reader more effectively through the authors’ interpretations and conclusions.
Thanks for this very relevant comment. We have structured the last two sections with subtitles. This revision does also solve of the following good comments about clarity and in-depth discussions.

18. While some prior studies are referenced (e.g., McClure et al., 2004; Plassmann et al., 2008), the discussion largely operates in isolation from existing literature. A more robust integration of the current findings with the relevant literature would help situate the study within a broader context, allowing for a deeper understanding of how these results contribute to existing knowledge in the field.
Thanks – we have made a total revision of the last sections and added references to the first part of the manuscript.

19.Although the discussion hints at cultural differences in consumer behavior between Moroccan and Tunisian participants, it does not delve deeply into the implications of these differences. A more thorough examination of how cultural perceptions and values influence preferences and WTP can provide valuable insights.
Thanks for the comment. Please see our previous responses.

20. The discussion states that "increased visual attention to the front of the package negatively impacted WTP," which is somewhat counterintuitive without a clear explanation. More attention should be paid to explaining why increased attention might signify confusion or uncertainty rather than value. A clear rationale for these findings would enhance the interpretative depth of the discussion.
Thanks for the comment. We have clarified this.

21.While the conclusion restates some key findings, it does not effectively summarize the main contributions of the study in a clear and concise manner. A more explicit presentation of the primary findings and their significance would help reinforce this study's contributions to the field.
Thanks – as mentioned above we have revised these last sections of the manuscript.

22. The conclusion briefly touches on cultural and demographic factors that may have influenced the findings but does not provide sufficient exploration of how these factors could be systematically examined in future research. Further elaboration of specific cultural influences or demographic characteristics would enhance the depth of the discussion.
Thanks for the comment. Please see our previous responses.

23.While the conclusion acknowledges challenges in replicating the results, it does not explicitly outline the limitations of the current study. Identifying and discussing the strengths and weaknesses of the research design, methodological choices, or sample characteristics would provide a more balanced view and increase the credibility of the findings.
Thanks for the comment. Limitations in relation to this study are added to the section.

24. Although this conclusion calls for additional research to explore unexplained variations in visual attention, it lacks specificity regarding what future studies might investigate. Outlining potential research questions, methodologies, or contexts that could build on these findings would provide clearer directions for subsequent research.
Thanks for the comment. We have structured the conclusion better and outlined potential research.

Reviewer 2 Report

Comments and Suggestions for Authors

The reasons for choosing olive oil seem to be insufficiently clear. The author wrote, "Olive oil was chosen as the object for the study as it entailed the dilemma between taste and information given and enabled the combination of variation in taste, health information, and information about region of origin." Could you provide more compelling evidence to support the representativeness of olive oil as a food product and offer a more reasonable explanation for choosing it as a study sample? Additionally, does the conclusion drawn in this study apply to other foods?

Polyphenols is a term relatively unfamiliar to the average consumer. Did the respondents in this study understand the properties and significance of this substance? My point is that their responses might differ depending on whether they are aware of it or not. In one scenario, their judgment might be based on scientific rationality, with the text content serving as a prompt. In another scenario, polyphenols could become a symbolic representation of health, leading consumers to develop a reverence for the unknown. Therefore, I believe it is necessary to delve into this potential mechanism.

Please provide the full term before using any abbreviations and thoroughly check the entire text. For instance, the full term for "ls" in Table 2 is unclear.

I seem to have missed the score range used in the evaluation process. While some numerical values are provided in the tables, it is unclear what the maximum score is and what the evaluation mechanism entails. I suggest adding some explanations to help readers better understand the source of the data, the methods of data collection, etc.

The discussion section could benefit from fewer paragraphs and by combining related conclusions into a cohesive argument. Besides this, you should engage in more in-depth critical analysis rather than simply stating the surface results of the study. Try to delve deeper into the theoretical and practical significance of the findings.

Some of the references are somewhat outdated; I suggest citing literature from the last five years only.

Author Response

1. The reasons for choosing olive oil seem to be insufficiently clear. The author wrote, "Olive oil was chosen as the object for the study as it entailed the dilemma between taste and information given and enabled the combination of variation in taste, health information, and information about region of origin." Could you provide more compelling evidence to support the representativeness of olive oil as a food product and offer a more reasonable explanation for choosing it as a study sample? Additionally, does the conclusion drawn in this study apply to other foods?
Thanks for the comment. We have outlined the choice for olive oil better, and in the discussion and call for further research we elaborate on benefits and limitation in our findings. 

2. Polyphenols is a term relatively unfamiliar to the average consumer. Did the respondents in this study understand the properties and significance of this substance? My point is that their responses might differ depending on whether they are aware of it or not. In one scenario, their judgment might be based on scientific rationality, with the text content serving as a prompt. In another scenario, polyphenols could become a symbolic representation of health, leading consumers to develop a reverence for the unknown. Therefore, I believe it is necessary to delve into this potential mechanism.
Thanks for your comment. Yes, the participants might have made their evaluation on different rationalities. The evaluation in phase one, where they only tasted the oil and without any information about the oil, would be different as in phase four, where the participants have been informed about origin, polyphenol, and healthiness. We have outlined this increase in information better in the manuscript and we also outline how the participants were informed about the polyphenol.

3. Please provide the full term before using any abbreviations and thoroughly check the entire text. For instance, the full term for "ls" in Table 2 is unclear.
Thanks for making us aware of this. We have checked the manuscript. 

4. I seem to have missed the score range used in the evaluation process. While some numerical values are provided in the tables, it is unclear what the maximum score is and what the evaluation mechanism entails. I suggest adding some explanations to help readers better understand the source of the data, the methods of data collection, etc.
Thanks for your comment. The WTP score had no limit as the participants would indicate the number/price they were willing to pay. We have made this clearer in the description of the setup. 

5. The discussion section could benefit from fewer paragraphs and by combining related conclusions into a cohesive argument. Besides this, you should engage in more in-depth critical analysis rather than simply stating the surface results of the study. Try to delve deeper into the theoretical and practical significance of the findings.
Thanks for this very relevant comment. We have structured the last two sections with subtitles and this revision provides both clarity and an in-depth discussion.

6. Some of the references are somewhat outdated; I suggest citing literature from the last five years only.
Thanks the comment. We have revised the reference list, although we cannot limit it to only recent published papers.

Round 2

Reviewer 1 Report

Comments and Suggestions for Authors

Dear authors,

Thank you for your comprehensive response to my review comments and for addressing the suggestions provided. I have carefully reviewed the revisions and am pleased to see that my concerns have been satisfactorily addressed. I believe the manuscript has been significantly improved and is now suitable for publication.

Reviewer 2 Report

Comments and Suggestions for Authors

The author has made excellent revisions to the manuscript, and I have no further suggestions.